# Anatomy-compliant medical image synthesis by latent diffusion models

**Nuno Capitão**[1]          N.M.nunomiguelFerreiraCapitao@tudelft.nl
**Yidong Zhao**[1]          Y.Zhao-8@tudelft.nl
**Yi Zhang**[1]          Y.Zhang-43@tudelft.nl
**Nathan Geerts**[1]          N.Geerts@student.tudelft.nl
**João V. Lopes**[2]          jlopes@fc.up.pt
**Qian Tao**[*1]          Q.Tao@tudelft.nl

[1] *Delft University of Technology, Lorentzweg 1, Delft, The Netherlands*

[2] *Department of Physics and Astronomy, Faculty of Sciences, University of Porto, 4169-007 Porto, Portugal*

## Abstract

Data scarcity presents a significant challenge for achieving optimal model performance in medical imaging, due to the limited availability of high-quality data. One potential solution to address this issue is to synthesize medical images using powerful generative models with conditioning prior. However, obtaining full anatomical annotations of all organs for anatomical conditioning is impractical, resulting in synthetic images with incoherent or hallucinated anatomy. In this paper, we propose an innovative medical image generation method based on state-of-the-art latent diffusion models (LDM). To tackle the anatomy compliance challenge, we leverage both the anatomical mask, which is specific to the organ of interest, and the edge information, which is general and easy to compute in the full field of view (FOV), as dual conditioning. Our method does not require extra annotations to achieve anatomy compliance. Our method was evaluated on the ACDC dataset and compared with GAN baselines. Results demonstrate that incorporating edge-based conditioning strongly complements image semantics, leading to high-quality, anatomy-compliant medical image generation.

**Keywords:** Latent diffusion models (LDM), Medical image synthesis, Edge.

## 1. Introduction

Learning-based approaches have gained prominence in many medical image processing tasks (Zhou et al., 2021). However, they rely heavily on high-quality labeled data, which is frequently unavailable given the high acquisition costs, rare pathologies, privacy concerns, and lack of annotation expertise (Castro et al., 2020). Generative AI offers a promising solution to the data scarcity problem (Dorjsembe et al., 2024). Previous medical image synthesis studies predominantly leveraged the SPADE (Park et al., 2019) framework which translates the semantic map to an image with Generative Adversarial Networks (GAN) (Goodfellow et al., 2014). Built upon SPADE, (Abbasi-Sureshjani et al., 2020) proposed generating cardiac magnetic resonance (CMR) images given heart segmentation maps. However, GAN-based generation can have degenerated quality (Müller-Franzes et al., 2023; Skandarani et al., 2021) and has recently been outperformed by diffusion-based generation (Ho et al., 2020; Dhariwal and Nichol, 2021; Rombach et al., 2022). Additionally and importantly, it

is often overlooked that the available semantic maps usually only cover specific organs (i.e. heart) (Abbasi-Sureshjani et al., 2020), leading to spurious and hallucinated generation of surrounding organs in the thorax and abdomen (i.e. lung, liver).

Image edges can delineate anatomical structures with little computational effort and can be used to guide image generation with GANs (Luo et al., 2021; Yu et al., 2019). Nevertheless, its use has not been explored in state-of-the-art diffusion models which promise superior quality to GANs. In this paper, we propose a novel LDM-based generative model conditioned on both incomplete organ annotations and image edges in the full field of view as dual anatomy conditioning. Our method ensures anatomical compliance in medical image generation, which has been understudied but is important for downstream tasks such as training AI models or educating radiologists (Skandarani et al., 2023).

## 2. Methods and Materials

We propose an LDM-based conditional generator $\mathcal{G}$ (Rombach et al., 2022) to translate anatomy into realistic images, which performs the diffusion process in an auto-encoded (AE) latent space. The anatomy is modeled by combining the partly labeled tissue map $y$ and the image edge $e_x$. The edge $e$ of an image $x$ is retrieved by Canny detector (Canny, 2009). The conditions are integrated into $\mathcal{G}$ through a binary boundary map derived from the image gradients, concatenated with a one-hot tensor representation of the semantic labels map, as represented in Figure 1.

We trained our models using the publicly available Automated Cardiac Diagnosis Challenge (ACDC) dataset (Bernard et al., 2018), and compared the proposed model to GAN-based conditional generation with SPADE (Park et al., 2019). To validate the utility of the edge as an additional condition, we assessed both models' performance with and without edge guidance, and evaluate quantitatively using metrics of FID, SSIM, NMSE. Additionally, we compared the Fourier space of generated images by different models.

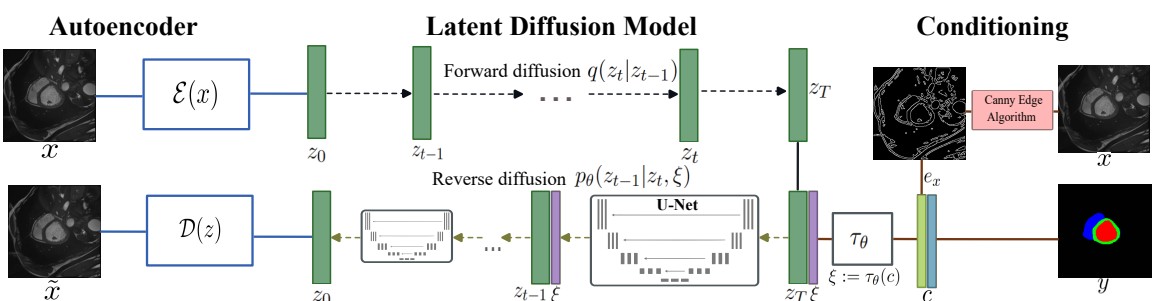

Figure 1: Illustration of the LDM-based edge conditioning pipeline.

## 3. Results and Conclusion

The quantitative evaluation results are reported in Table 1. The proposed LDM-based generation achieved the best performance in terms of FID (61.9/98.1) compared to the best SPADE variant (101.9/161.4), with or without edge guidance. Our LDM has a lower SSIM

than the SPADE-based model without edge guidance. However, the edge-guided LDM achieved an SSIM of 0.608 which is much higher than the SPADE variant (0.556). Using edge guidance also significantly improved the generation quality for SPADE variants.

Figure 2 presents some examples of generated CMR images and their Fourier frequency spectra. For LDM, we observe that the generated images are closest to the ground truth and correctly delineate organ details that the other models fail to capture. Nevertheless, we note that all edge-informed models show similar improvement. Moreover, we observe that for the edge-informed models, LDM aligns closely with the ground truth in both low and high-frequency ranges, whereas SPADE models only achieve closer alignment in higher frequencies.

Table 1: Image generation performance measured by FID, SSIM and NMSE.

| Method | w/ edge conditioning | | | w/o edge conditioning | | |
| | FID ↓ | SSIM ↑ | NMSE ↓ | FID ↓ | SSIM ↑ | NMSE ↓ |
| --- | --- | --- | --- | --- | --- | --- |
| LDM | 61.9 | $0.608 \pm 0.123$ | $0.071 \pm 0.035$ | 98.1 | $0.283 \pm 0.094$ | $0.136 \pm 0.080$ |
| SPADE | 109.8 | $0.556 \pm 0.119$ | $0.117 \pm 0.041$ | 160.3 | $0.328 \pm 0.093$ | $0.132 \pm 0.077$ |
| SPADE w/ VAE | 101.9 | $0.556 \pm 0.128$ | $0.104 \pm 0.034$ | 161.4 | $0.338 \pm 0.090$ | $0.145 \pm 0.093$ |

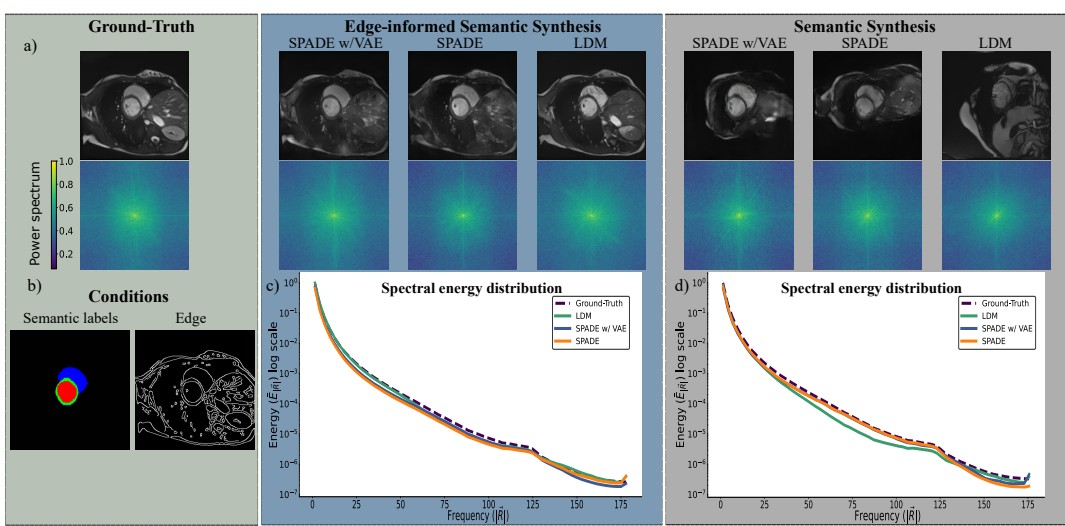

Figure 2: Qualitative results: (a) The ground truth and its Fourier spectrum. (b) Left: Segmentation map. Right: Edge anatomical boundaries by the Canny operator. (c),(d) Comparison of the average model spectra as a function of $|\overrightarrow{\mathbf{R}}| = \sqrt{x^2 + y^2}$, representing a radius distance from the 2D grid origin.

In conclusion, we proposed a novel medical image generation method using LDM with dual conditioning by both semantic labels and Canny edges. Our experiments showed that our method improved image synthesis quality and anatomy compliance, without requiring additional exhaustive annotations. Furthermore, we have shown that LDM outperforms GAN-based models in both the image and frequency space.

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
