# OpenReview forum: "Anatomy-compliant medical image synthesis by latent diffusion models"
_MIDL.io/2024/Short_Papers — MIDL 2024 Short Papers_

### Official Review · Reviewer_Tur8 · 2024-04-23

**Confidence:** 4
**Final Rating:** 3.5

**Review:**

Key idea: The authors propose latent diffusion models to synthesize images conditioned on semantic labels and edges.  The proposed approach is demonstrated on cardiac MR (CMR) data where CMR images are synthesized conditioned on segmentation masks and canny edges to generate cardiac MR images.

 1) Quality
Overall, the problem is well-defined and described, and the paper is easy to follow. The novelty of the proposed approach is clear and informative illustrations are provided. Spectral energy distribution of the synthesized samples is also used as a metric.
However, the real-life application is not clear. The approach assumes availability of edges of the desired anatomy. Furthermore, it is not shown how the model would perform if only a subset of edges is provided as conditions.

2) Clarity:
The paper is reasonably easy to understand structure.

3) Originality
The paper contains dual conditions ( edge and segmentation mask in LDM)

4) Significance
The proposed approach seems to solve the issue of lacking annotation masks for all organs and outperform other models.
However, the real-life application is unclear.

---

### Decision · Program_Chairs · 2024-04-26

Accept